# Drivers and pressures behind insect decline in Central and Western Europe based on long-term monitoring data

Quintana Rumohr[1], Christian Ulrich Baden[2]\*, Matthias Bergtold[3], Michael Thomas Marx[2], Johanna Oellers[1], Michael Schade[4], Andreas Toschki[1], Christian Maus[2]

1 gaiac, Research Institute for Ecosystem Analysis and Assessment, Aachen, Germany, 2 Bayer AG, Monheim am Rhein, Germany, 3 BASF SE, Limburgerhof, Germany, 4 Syngenta Crop Protection, Basel, Switzerland

\* christian.baden@bayer.com

**Data Availability Statement:** All relevant data are within the paper and its Supporting Information files.

## Abstract

Insect declines have been discussed intensively among experts, policymakers, and the public. Albeit, decreasing trends have been reported for a long time for various regions in Europe and North America, but the controversial discussion over the role of specific drivers and pressures still remains. A reason for these uncertainties lies within the complex networks of inter-dependent biotic and abiotic factors as well as anthropogenic activities that influence habitats, communities, populations, and individual organisms. Many recent publications aim to identify both the extent of the observed declines and potential drivers. With this literature analysis, we provide an overview of the drivers and pressures and their inter-relationships, which were concluded in the scientific literature, using some of the best-studied insect groups as examples. We conducted a detailed literature evaluation of publications on Carabidae (Coleoptera) and Lepidoptera trends with data for at least 6 years in countries of Central and Western Europe, with a focus on agricultural landscapes. From the 82 publications identified as relevant, we extracted all reported trends and classified the respective factors described according to the DPSIR model. Further, we analysed the level of scientific verification (presumed vs correlated vs examined) within these papers for these cited stressors. The extracted trends for both species groups underline the reported overall declining trend. Whether negative or positive trends were reported in the papers, our semi-quantitative analysis shows that changes in insect populations are primarily anthropogenically driven by agriculture, climate change, nature conservation activities, urbanisation, and other anthropogenic activities. Most of the identified pressures were found to act on habitat level, only a fraction attributed to direct effects to the insects. While our analysis gives an overview of existing research concerning abundance and biodiversity trends of carabids and lepidopterans, it also shows gaps in scientific data in this area, in particular in monitoring the pressures along with the monitoring of abundance trends. The scientific basis for assessing biodiversity changes in the landscape is essential to help all stakeholders involved to shape, e.g. agriculture and other human activities, in a more sustainable way, balancing human needs such as food production with conservation of nature.

**Funding:** CUB, MTM, CM, MB, and MS are co-authors and contributed to the establishment of the study concept and to the evaluation and interpretation of the data, and to the preparation of the manuscript, as stated. The study was financially supported by Bayer, BASF, and Syngenta.

**Competing interests:** The authors have declared that no competing insterests exist.

## Introduction

The decline in insect diversity and abundance is currently attracting the attention of scientists, politics, and the general public, especially after recent publications [1–7] revealed significant declines which were widely reported in the media [8].

This decline has been known to entomological and ecological science for a longer time [9–13], however, since not all data sets show consistently declining trends [6, 14–18], there has been a controversial discussion of the significance and the extent of the declines, especially when considering local, regional or global scales [19]. Still, the public awareness in the last five years has focussed attention on this topic and created a persistent pressure on politicians to address it [20]. In order to effectively use the momentum of public attention to promote policies to preserve, restore and promote biodiversity e.g. in sustainable agricultural production, it is important to target the relevant policies. So far the general policy toolbox is implemented in the Common Agricultural Policy including agricultural support policies and international trade negotiations [8, 21]. Some authors state that we already know enough to act in order to halt or reverse insect decline [22], whereas others suggest that investment in basic science is warranted to better understand the phenomenon [23–25], to inform more targeted and effective countermeasures. However, both positions are not necessarily in contradiction, since research into the causes of insect decline does not preclude action where this is already possible and should not be seen as a justification to delay action [26, 27]. Being aware of this, many of the recent publications aim to identify the potential causes behind these trends [8, 28–30]. However, while many causes, e.g. agricultural intensification, have been well described in general, we only have limited information about the complex interplay of stressors in detail. Due to different approaches and some shortcomings, evaluating the available information has led to controversial discussions about the relevance of specific drivers [24, 31–34].

One of the reasons for this is, in itself, a complex issue, primarily due to the problem that insect population abundance can—but diversity cannot—be encompassed easily as a single number, or be compared by indices alone, e.g. more/better, less/worse. Even though it has been attempted, diversity is a habitat-specific, qualitative value and it also depends on specific traits of species groups (e.g. r/K selection strategy). While the populations of species can decrease over time in response to stressors, populations of other species may increase at the same time due to the same stressors, e.g. by reduction of inter-species competition. A change to better habitat quality, for example, could be accompanied by a decreasing overall abundance whilst an increase of specific characteristic taxa is observed. Nevertheless, negative trends can be observed in many places, i.e. the abundance of insects is decreasing in specific areas, e.g. in nature conservation areas or the agricultural landscape [2, 29]. Consequently, the discussion on appropriate policy responses to this reflects the underlying complexity and inter-relationship between these pressures, which is typical of multi-trophic ecosystems with multiple anthropogenic influences [22, 26]. In a recently published study, a questionnaire among experts was conducted to analyse population trends, threats and conservation measures for insects [32]. Going a step beyond this approach of eliciting expert opinions, the aim of our study was to extract and analyse causes that authors examined and discussed in their publications on long-term population and biodiversity trends of carabids and lepidopterans in the agricultural landscape of Central and Western Europe.

The objectives of this analysis are

a. to compile publications containing long-term datasets for selected insect taxa (Carabidae and Lepidoptera) in agricultural landscapes of a defined region

b. to extract and summarise the observed long-term population and biodiversity trends along-side with the pressures concluded and postulated

c. to provide an overview and analysis of the pressures extracted and their drivers, which might be influencing insect abundance and diversity

d. to identify knowledge gaps and future research needs.

## Methods and data basis

The review process went through following steps: literature search, descriptive analysis of content, extraction of trends and pressures, input in a database, classification of trends and pressures, and visualisation.

## Literature search

A systematic and standardised literature search and analysis approach was developed to evaluate published long-term data in peer-reviewed and 'grey' literature covering different insect groups, particularly in the agricultural landscape of countries in Central and Western Europe: i.e. Austria, Belgium, Germany, Luxembourg, Switzerland, the Netherlands, and the United Kingdom. This geographic scope was chosen because it covers a structurally relative comparable environment with landscape structures, with landscapes significantly shaped by human activities, especially agriculture, and where an almost 300-year long tradition of entomofaunistics entailed a relatively good availability of long-term insect distribution data, so that we could expect a sufficient number of suitable data sources published in English or German.

In a first step, different categories and lists of keywords were defined to ensure maximal efficiency of the search and consistency of the search results. The following four keyword categories were applied: taxonomic group, time-based/methodical context, measured parameter, and land use/landscape context. S1 Table shows those four categories with the keywords used in English and German. The first keyword category contains the scientific and common names of the insect group. The second category includes the terminology of time-based studies and the methodical context. The third category consists of the terminology considering the measured parameter, and the fourth category describes the landscape and land use context. Combinations of three to four keywords of the four developed categories were used for searches in English and German on the following search platforms: Google Scholar, Web of Science, ScienceDirect and Google.

In a second step, the preliminary search results were quality checked and filtered. We excluded publications on individual species as these studies often addressed very specific research questions and did not fit our broader scope. In addition, although we tried to find studies with long-term data, there is no uniform definition of the time frame 'long-term'. This was particularly important because short-term changes can be seen as fluctuations rather than trends. Therefore, we defined long-term data here as surveys with data collected over a period of at least six (not necessarily consecutive) years. This was a compromise to include as many publications as possible while still allowing for a reasonable long time period.

Finally, the preliminary search revealed the most publications with long-term data for the insect groups of Lepidoptera and Carabidae. Therefore, we selected the remaining publications of these two groups for the exemplary evaluation in this analysis.

## Trend and data evaluation

Each publication was reviewed in different steps. As many publications yielded more than one trend, in a first step, every trend of a defined group (e.g. Lepidoptera or Carabidae in general,

generalists, specialists or grassland species) was extracted, analysed and assessed individually. As with the publications, trends of individual species were excluded. The extracted trends were classified into five different trend types: increase, decrease, stable, shift (e.g. changing species composition) or inconsistent (change of the measured parameter was observed, but without consistent direction of the change). In a second step, each extracted trend was characterised regarding the following subjects:

- **Taxonomic identification level:** insects were determined with varying degrees of precision, i.e. species, family, suborder, order level

- **Location:** detailed description of the sampling area of each trend:

  ○ the exact location of the study area was recorded (country, locality, coordinates)

  ○ a location could be a single sampling site as well as a larger area, a habitat type in a defined region or one to several countries, so it was classified as a local, regional, national or multi-national study area

- **Sampling method:** e.g. pitfall traps, flight intersection traps, light traps, transect counts

- **Measured parameter:** e.g. number of individuals, biomass, number of species, distribution

- **Parameter unit:** for a further description of the parameters, the parameter unit, such as occupied grid cells, counts or alpha-diversity, was added

- **Timespan:** the recording type (continuous, interval, scattered), as well as the start and end years of the study/intervals, were noted

- **Additional information:** tags, a short description of the publication or study design

In order to classify, structure and record the extracted trends and metadata, they were compiled in a database for further analysis of the trends.

## Evaluation of population trends

**Drivers and pressures of the extracted trends.**   In our analysis, we used an adapted DPSIR model (drivers, pressures, states, impacts and responses [35], see Fig 1) to disentangle the complex network of inter-dependent biotic and abiotic factors as well as anthropogenic activities that influence habitats, communities, populations and individual organisms. In this conceptual model, the network is split into five levels. Drivers, for example agricultural intensification or nature conservation activities, lead to multiple pressures (e.g. increasing or decreasing use of fertilisers), which influence one or several states, such as habitats and organisms. The influences are shown in the model as impacts, such as a reduced availability of habitats or resilience of populations. These impacts induce a response, which in turn affect the drivers, pressures, states and impacts. In our evaluation, we adapted the model to focus only on drivers, pressures, and state, and excluded impacts and responses because they were not the central subject of the evaluated publications. In many cases, the studies did not provide the information necessary to include these latter parameters.

The compiled trends were further evaluated by extracting every cause that was explicitly mentioned by the authors of the evaluated studies. Such causes were recorded as pressures on a detailed, low aggregation level. In total, 87 specific, distinct pressures were extracted as a basis for the following categorisation into pressure classes, their drivers and states according to the DPSIR model (compare the full list of detailed pressures with categories S3 Table and definition of categorisation S2 Table):

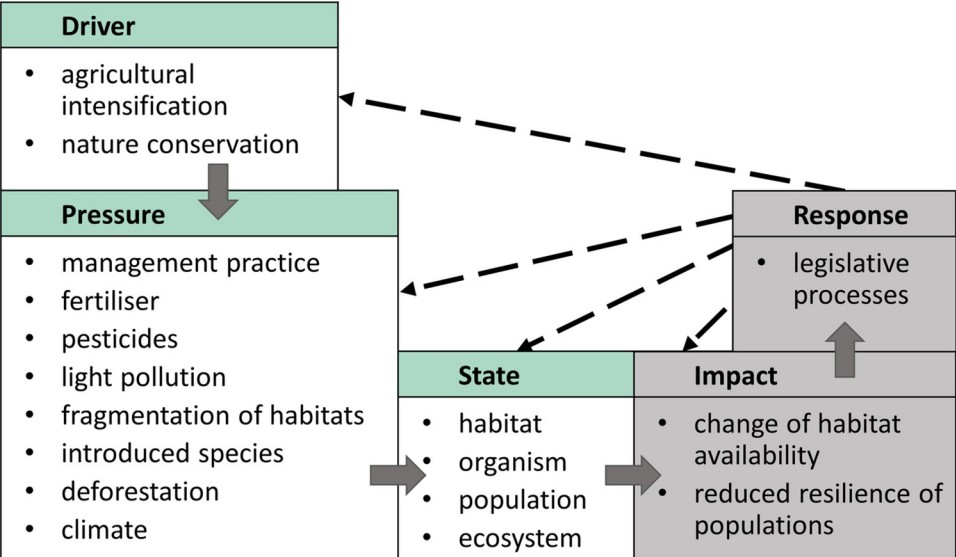

**Fig 1. DPSIR model with examples.** (drivers, pressures, states, impacts, responses; modified after EEA 1999 [35]; shaded in grey: not considered in this publication).

- **pressures:** specific, distinct stressors that were mentioned as the causes behind the described trends, gathered from the evaluated publications.

- **pressure classes:** aggregation of pressures into 14 classes marking broader topics; e.g. 'agri-environment schemes or measures', 'mowing regime', 'drainage' and 'habitat restoration' were aggregated among others in the pressure class 'management practices'.

- **drivers:** forces behind the pressures aggregated in 11 classes; each pressure is related to a specific driver independently from its pressure class; e.g. the drivers nature conservation and agricultural intensification can both result in a change of management practices.

- **states:** describing the point of influence of the respective pressures; grouped in 4 classes; e.g. the pressure 'isolation/fragmentation' changes the habitat availability, while the 'mowing regime' can have an impact on both, habitat (condition/availability) and inhabiting organisms directly.

If a publication did not include detailed information on a concluded pressure, e.g. 'perturbation in general', or the impact was specific but the driver was not mentioned, e.g. 'change of vegetation', we used categories like 'various pressures' or 'various drivers' for the aggregation. Additionally, it was recorded if the evaluated publications did not give any information about possible pressures related to the respective trends.

**Scientific verification of the pressures.** We classified the pressures described by the authors of the evaluated publications into four different scientific verification levels:

- **pressure examined and statistically evaluated:** The cause behind a trend was directly assessed and considered in the study design as well as statistically evaluated.

- **pressure examined:** The cause behind a trend was directly assessed and considered in the study design, but not statistically evaluated.

- **pressure correlated:** The cause behind a trend was not directly assessed and considered in the study design; however, the authors of the evaluated publication correlated the cause

behind the trend using different information like the autecology of the species or data from other sources (e.g. weather/climate data, land use indicators).

- **pressure presumed:** The cause behind a trend was not directly assessed and considered in the study design; the authors of the evaluated publication hypothesised that a certain pressure was the cause, based on their knowledge (e.g. about the changes of habitats in the past and the ecological requirements of the considered taxonomic group or species, or about results of previous studies on comparable topics).

**Pressure rating.** All pressures were classified either as 'important' or 'not important' as identified by the authors of the paper. If the authors mentioned a pressure as an underlying cause of a trend, this pressure was categorised as important. If the authors explicitly discussed a pressure as not important, it was categorised accordingly in this analysis.

**Summarising analysis and presentation of the trends and the underlying causes.** The focus of the analyses was on the decreasing and increasing trends for both taxonomic groups; they contained:

- the evaluation of specific pressures mentioned by the authors of the publication in combination with their categorisation in pressure classes, drivers and states according to the DPSIR model,

- a visualisation of the extracted pressures in alluvial diagrams.

In the alluvial diagrams, the relations of the drivers, pressure classes and states are depicted. The width of the bars depends on the number of all underlying, specific pressures mentioned by the authors for the different trends. In this way, a single pressure may be included several times, depending on the frequency of its reference in the publications. For the visualisation, the specific pressures were summarised in the described pressure classes (see above). The alluvial diagrams were created using the application 'RAWGraphs' [36].

## Results

### Overview of the analysed publications

In total, 82 publications matched our search criteria (28 for carabids, 49 for lepidopterans and five publications containing both groups, S4 Table). The majority of study areas were located in the United Kingdom (29 publications), Germany (25 publications) and the Netherlands (16 publications) (for detailed numbers of studies per country, see S1A Fig). A few publications analysed data consisting of multi-national datasets. The data in these publications comprises periods between 1750 AD for lepidopterans [37], 1840 AD for carabids [38, 39] and the late 2010s e.g. [40–45] (S1A–S1C Fig).

### Overview of the data basis

Table 1 shows the results of the literature analysis for trends and pressures. In some publications, several different trend types were identified and discussed, which results, when calculated separately, in a higher sum than the total number of publications. From these publications, 104 and 187 trends for Carabidae (C.) and Lepidoptera (L.) were extracted in total. Within the 33 carabid publications in total, there were 21 with at least one decreasing, 15 with at least one stable, and 19 with at least one increasing trend, seven with descriptions of shifts and six with inconsistent trends. Within the 54 Lepidoptera publications in total, there were 45 with at least one decreasing, 25 with at least one stable, and 30 with at least one increasing trend, six descriptions of shifts and five inconsistent trends. Thus, for both groups,

**Table 1. Numbers of analysed publications and of trends and pressures.** Described in the analysed publications, with indication whether the pressures were discussed and which level of importance was assigned to them in the original publications; *multiple trend types per publication possible.

| | Carabidae | | | | | | Lepidoptera | | | | | |
|---|---|---|---|---|---|---|---|---|---|---|---|---|
| | decrease | stable | increase | shift | inconsistent | total | decrease | stable | increase | Shift | inconsistent | total |
| publications* | 21 | 15 | 19 | 7 | 6 | 33 | 45 | 25 | 30 | 6 | 5 | 54 |
| extracted trends | 41 | 23 | 25 | 7 | 8 | 104 | 90 | 36 | 48 | 7 | 6 | 187 |
| trends *with* pressures discussed | 28 | 1 | 18 | 7 | 3 | 57 | 74 | 14 | 36 | 6 | 3 | 133 |
| trends *without* pressures discussed | 13 | 22 | 7 | 0 | 5 | 47 | 16 | 22 | 12 | 1 | 3 | 54 |
| pressures discussed in total | 84 | 1 | 28 | 12 | 5 | 130 | 394 | 24 | 71 | 16 | 4 | 509 |
| *important* pressures discussed in total | 81 | 1 | 26 | 11 | 3 | 122 | 389 | 24 | 70 | 16 | 4 | 503 |

most trends were decreasing (39% C. and 48% L.), followed by increasing (24% C. and 26% L.) and stable (22% C. and 19% L.) trends. Shifts and inconsistent trends play only a minor role with less than 10% for both taxa.

When considering the different parameters measured within the trend types, there are differences between the two taxa (Table 2). For carabids, the number of individuals is the dominant parameter reported with decreases (18x), increases (11x) and stable trends (10x). The next most frequent parameter is the species richness (9 decreases, 4 increases and 7 stable) followed by distribution (5 decreases, 5 increases and 2 stable). Lepidopteran decreases are reported with different parameters (28 abundance by indices, 21 distribution, 20 species richness and 11 number of individuals) (note that the differentiation between "abundance by indices" and "number of individuals" is made since some studies report abundances not by individual numbers, but by means of indices like TRIM (TRends and Indices for Monitoring data, [46] or RRR (Relative Reporting Rate)). The dominant parameters for increases and stable trends are the distribution (16 increase and 11 stable) and abundance (15 increase and 13 stable).

The authors of the evaluated publications mentioned at least one pressure for the majority of reported trends (55% C. and 71% L.). When those trends were excluded for which no pressure was discussed, for Carabidae were on average 2.3 pressures (130/57), and for Lepidoptera on average 3.8 pressures (509/133) discussed per trend. Most of these pressures were

**Table 2. The number of trends separated into the different parameters for Carabidae and Lepidoptera.**

| Parameter | Carabidae | | | | | | Lepidoptera | | | | | |
|---|---|---|---|---|---|---|---|---|---|---|---|---|
| | decrease | stable | increase | shift | inconsistent | total | Decrease | stable | increase | shift | inconsistent | total |
| abundance (only with indices) | 3 | 1 | 1 | 0 | 0 | 5 | 28 | 13 | 15 | 0 | 1 | 57 |
| Biomass | 1 | 2 | 0 | 0 | 0 | 3 | 4 | 2 | 4 | 0 | 0 | 10 |
| Distribution | 5 | 2 | 5 | 1 | 1 | 14 | 21 | 11 | 16 | 1 | 2 | 51 |
| diversity indices | 3 | 1 | 3 | 0 | 0 | 7 | 3 | 1 | 1 | 0 | 0 | 5 |
| dominance | 0 | 0 | 0 | 0 | 0 | 0 | 0 | 0 | 1 | 1 | 0 | 2 |
| evenness | 0 | 0 | 1 | 0 | 0 | 1 | 0 | 0 | 0 | 0 | 1 | 1 |
| frequency | 0 | 0 | 0 | 0 | 0 | 0 | 1 | 0 | 0 | 0 | 0 | 1 |
| functional diversity | 1 | 0 | 0 | 0 | 0 | 1 | 0 | 0 | 0 | 0 | 0 | 0 |
| number of individuals | 18 | 10 | 11 | 0 | 2 | 41 | 11 | 7 | 5 | 0 | 0 | 23 |
| phylogenetic diversity | 1 | 0 | 0 | 0 | 0 | 1 | 0 | 0 | 0 | 0 | 0 | 0 |
| species composition | 0 | 0 | 0 | 6 | 2 | 8 | 2 | 0 | 2 | 5 | 0 | 9 |
| species richness | 9 | 7 | 4 | 0 | 3 | 23 | 20 | 2 | 4 | 0 | 2 | 28 |
| total | 41 | 23 | 25 | 7 | 8 | 104 | 90 | 36 | 48 | 7 | 6 | 187 |

considered important for the trend observed in both taxa groups. For both groups, the highest numbers of pressures were noted for decreasing trends (66% C., 77% L.) followed by increasing trends (21% C., 14% L.). For butterflies and moths, stable trends were mentioned with 24 out of 503 pressures (5%), whereas for carabids, only one stable trend with one pressure was identified.

## Applying the DPSIR model and analysing the complexity of causes of the trends

The following results analyse the frequency of pressures mentioned in publications and their classification according to the DPSIR model for the dominant trend types, i.e. decreases and increases. The extracted pressures and the frequency of their mention by the authors are presented in the S3 Table. Alluvial diagrams show the extracted pressures classified in pressure classes (middle section), their derived drivers (left section) and states (right section) (Part a of Figs 1–4). Additionally, Part b of Figs 1–4 gives the frequency of extracted pressures classified according to scientific verification levels and pressure classes.

## Overview of the findings

For Carabidae as well as for Lepidoptera, key drivers of decreasing trends were anthropogenic activities in general (32% / 35%), followed by agricultural intensification (23% / 24%). Climate change was indicated as a driver in 15% / 9% of the analysed cases. Important pressure classes for both groups were land use (25% / 24%), management practices (15% / 16%), and climate (17% / 13%). The vast majority of these pressures were classified to act on the habitat and its availability, or on habitats and organisms, only a small fraction (7% / 3%) on the organisms directly. In the majority of analysed studies, the pressures were presumed with a declining trend (69% / 63%), in by far smaller fractions correlated with the decline (15% / 27%) or evaluated in more detail (16% / 11%).

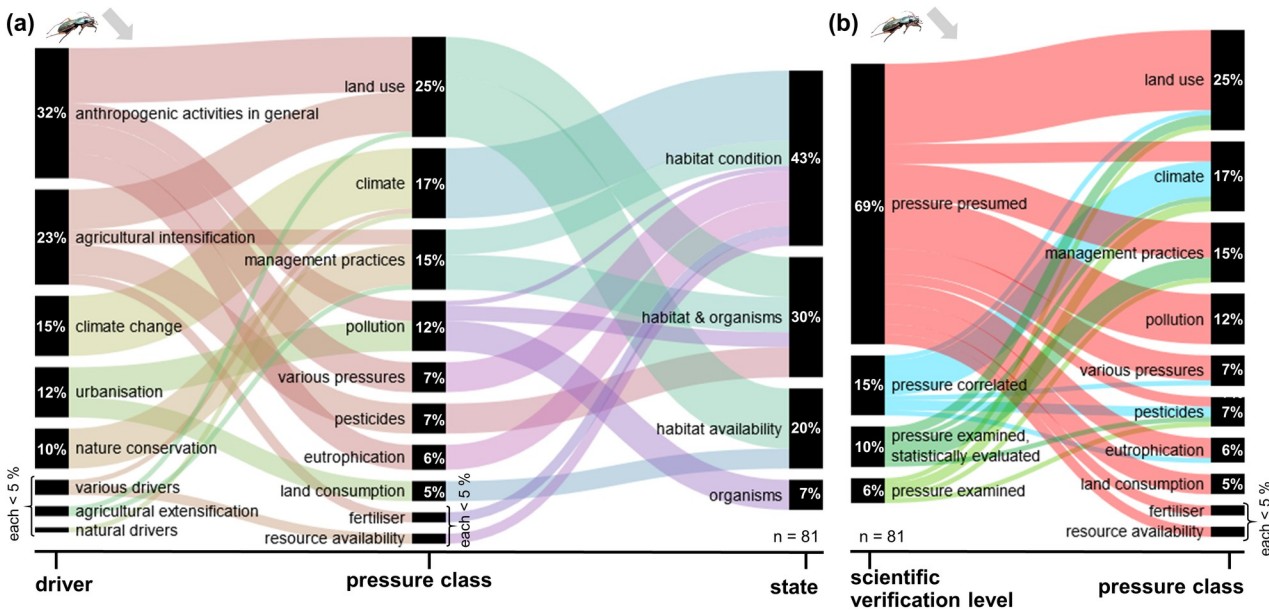

**Fig 2. Frequency of extracted pressures for decreasing carabid trends.** These trends are based on 81 extractions in total. Specific pressures classified into pressure classes and (a) their corresponding drivers and states, or (b) their scientific verification levels.

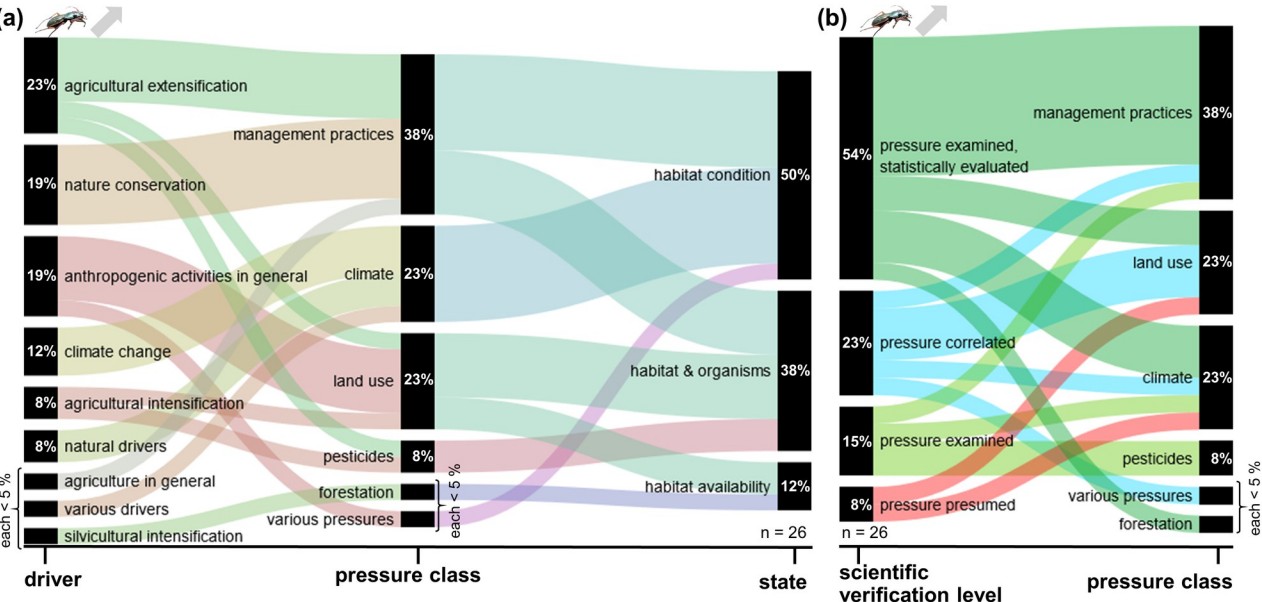

**Fig 3. Frequency of extracted pressures for increasing carabid trends.** These trends are based on 26 extractions in total. Specific pressures classified into pressure classes and (a) their corresponding drivers and states, or (b) their scientific verification levels.

The drivers behind increasing trends were more heterogeneous: for Carabidae, agricultural extensification (23%), nature conservation, anthropogenic activities in general (both 19%), and climate change (12%) were indicated as key drivers, for Lepidoptera climate change (33%), nature conservation (21%), anthropogenic activities in general (14%), and extensification in forestry (10%). Major pressure classes for both groups were climate (23% / 40%), management practices (38% / 13%), land use (23% / 14%), and, in the case of Lepidoptera, also

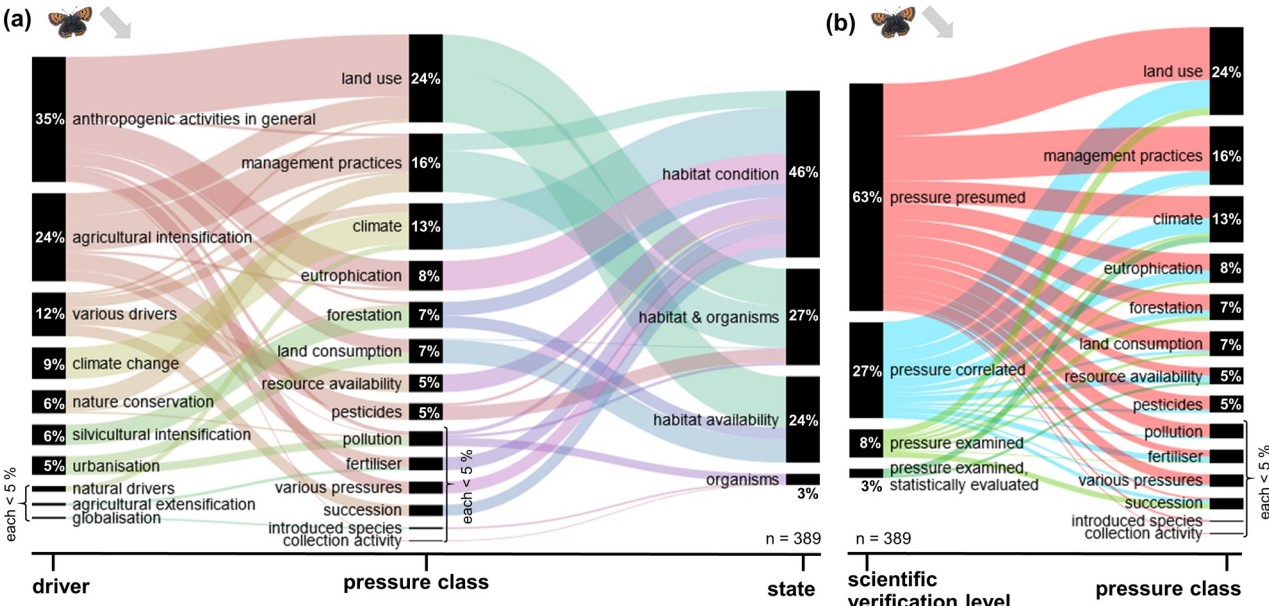

**Fig 4. Frequency of extracted pressures for decreasing Lepidoptera trends.** These trends are based on 389 extractions in total. Detailed pressures classified into pressure classes and (a) their corresponding driver and state, respectively, or (b) their scientific verification level.

forestation (14%). For both groups, there pressures acted mainly on the habitats and their availability (88% / 85%). Between both groups, there were substantial differences between the verification levels for the pressures: whereas in the case of Carabidae, 69% of the pressures were examined in detail, 23% correlated, and only 8% presumed, for Lepidoptera, 25% of the pressures were studied in detail, 29% correlated, and 46% presumed.

**Carabidae decreasing trends.**   In total, eight drivers could be identified, based on 81 important pressures indicated by the authors of the evaluated publications, that account for decreasing trends of carabids (Fig 2A). The most frequent driver is anthropogenic activity in general (32%). This category included the pressures mentioned by the authors of the analysed publications, e.g. industrial pollution, land use in general, acidification, which were either rarely mentioned or not further specified, and could therefore not be further grouped. Other, also specific anthropogenic activities, e.g. urbanisation and nature conservation, were mentioned several times and therefore grouped and listed separately. The next highest mentioned drivers were agricultural intensification (23%), climate change (15%), urbanisation (12%) and nature conservation activities (10%). The most frequently cited pressure class for this trend type was land use (25%), mainly in the context of anthropogenic activities in general and agricultural intensification, affecting habitat availability and habitats & organisms. This pressure class is followed by climate (17%), mainly due to climate change, which changes the habitat conditions. The influence on habitat conditions (43%) has the highest share due to different pressures followed by habitat & organisms (30%) and habitat availability (20%). A minor share of the pressures (7%) were cited to act on the affected organisms directly. Land use and climate change together with pressure classes management practice (15%), pollution (12%), various pressures (7%), pesticides (7%), eutrophication (6%) and land consumption (6%) accounted for the 95% of the cited pressures.

The scientific verification level associated with the pressures shows that the majority (69%) are classified as presumed by the authors, as they were not part of the methodological design or the evaluation process within the publication (Fig 2B). The remainder (31%) of the pressures are either analysed for correlation in the evaluated publication or explicitly examined with or without an accompanying statistical evaluation. The highest share of these (40%) dealt with climate as pressure class. Thus, the pressure class climate contains 71% (10/14) extracted pressures that were derived by correlation or explicitly examined. Although the overall number of times pesticides was explicitly cited as a pressure was relatively small (7%), a higher proportion of correlated and examined trends compared to presumed ones was found (67%).

**Carabidae increasing trends.**   Nine drivers could be identified for increasing trends of carabids (Fig 3A), based on 26 pressures indicated by the authors of the publications evaluated. The most frequent of these drivers was agricultural extensification (23%) followed by anthropogenic activities in general (19%), nature conservation (19%) and climate change (12%).

The pressure class most frequently cited for this trend type is land management practice (38%), mainly due to agricultural extensification and nature conservation activities affecting organisms as well as habitat conditions. Land use and climate are the next most cited pressure classes (both 23%). The former, driven by anthropogenic activities in general, has an impact on habitats & organisms and habitat availability. The latter changes habitat conditions under the influence of climate change and natural pressures. Half of the increasing trends of carabids can be traced back to a change of habitat conditions, 38% of the pressures affect both habitat & organisms, and 12% change the habitat availability. No pressures were found to be influencing the organisms directly. For two (8%) of the described increasing trends of carabids, the authors identified the reason as a change in pesticide usage.

In contrast to the decreasing trends, most of the pressures (92%) are explicitly examined or at least correlated within the publications (Fig 3B). More than half of the pressures (54%) are

supported by statistical analyses, and only a minor share of 8% of the pressures were presumed by the authors.

**Lepidoptera decreasing trends.** Twelve drivers were identified (Fig 4A), based on 389 pressures indicated by the authors of the evaluated publications for decreasing trends of Lepidoptera. The most frequent drivers are anthropogenic activities in general (33%), agricultural intensification (23%) and multiple drivers (12%), i.e. in those cases, the detailed pressure is allocated to a combination of several drivers. Further drivers are climate change (9%), nature conservation (6%), silvicultural intensification (6%) and urbanisation (5%).

In total, 14 pressure classes were cited for the decreasing Lepidoptera trends. The most frequently cited pressure class is land use (24%) driven by anthropogenic activities in general and agricultural intensification. It is followed by management practices (16%) influenced by agricultural intensification and nature conservation. Both pressure classes have direct effects on habitats & organisms, while land use additionally affects the change of the habitat availability. Other pressure classes with 5% or more are climate (13%), eutrophication (8%), forestation (7%), land consumption (7%) resource availability (5%), pesticides (5%). Most pressures are inducing changes in habitat conditions (46%). The second most common effect of pressures is on habitat & organisms (27%), followed by habitat availability (24%). Of the cited pressures 63% were presumed, 26% were derived by correlation, and 11% were explicitly examined within the publications (Fig 4B). The highest proportion of explicitly examined or correlated pressure classes were in those pressure classes that were rarely cited such as (vegetation) succession (83%, 10/12), resource availability (53%, 10/19) or fertilisers (50% 7/14). Out of the more commonly cited pressure classes, climate (52%, 26/50) and land use (40%, 38/95) were most often examined or correlated.

**Lepidoptera increasing trends.** For increasing trends of Lepidoptera, 8 drivers could be identified (Fig 5A), based on 70 pressures indicated by the authors of the evaluated publications. The most frequent driver is climate change (33%) followed by nature conservation (21%) and other anthropogenic activities (14%). The most frequently cited pressure class for this trend is the climate (40%) mainly due to climate change and natural pressures changing

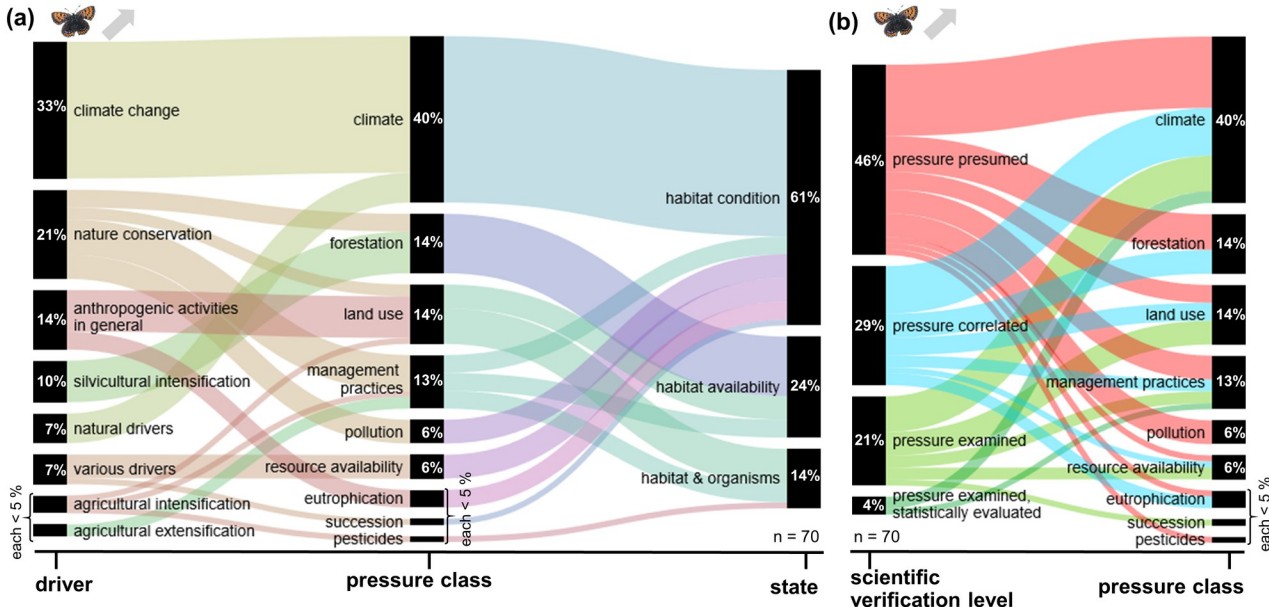

**Fig 5. Frequency of extracted pressures for increasing Lepidoptera trends.** These trends are based on 70 extractions in total. Specific pressures classified into pressure classes and (a) their corresponding drivers and states, or (b) their scientific verification levels.

habitat conditions. The two following pressure classes with a share of 14% each are land use and forestation, both changing the habitat availability whereas the former also affects habitats and organisms. Another pressure with a share of more than 13% is land management practice. It is driven by nature conservation activities and affects habitat condition and availability as well as the habitat and organisms directly. Of the pressures, 46% were presumed by the authors of the publications, 29% were derived by correlation, and 25% were explicitly examined within the publication either with or without statistical analyses (Fig 5B). Similar to the decreasing trends, the highest proportion of explicitly examined or correlated pressure classes were in those pressure classes that were rarely cited such as (vegetation) succession (100%, 1/1) or resource availability (70%, 7/10). Of the more frequently mentioned pressure classes, land use (40%, 38/95), climate (57%, 16/28) and management practices (56%, 5/9) were most often examined or correlated.

## Discussion

This analysis of long-term monitoring data on carabids and lepidopterans provides an overview of the evidence for and currently cited potential causes of population and biodiversity changes in two insect taxa in selected countries in Central and Western Europe, with a focus on agricultural landscapes.

### Parameters measured

Biodiversity is a complex phenomenon with many dimensions and facets, and accordingly, there is a broad variety of parameters related to the various dimensions and the overall phenomenon. Relevant parameters include for instance abundance, biomass, species richness, diversity, distribution, and others. All these parameters develop dynamically in insect populations and communities, under the influence of a broad spectrum of drivers. Frequently, the different parameters are interdependent and influence each other, where this can, but does not necessarily has to happen in the same direction; for example, abundance and biomass may be correlated in some (yet not all) cases, whereas there may on the other hand be instances where an increase of diversity may come along with a decrease of abundance or biomass, e.g. when a dominant species gets lost from a community. In our study, we have evaluated different parameters, according to which changing trends were described in the literature, in one integrative analysis. This has primarily practical reasons. First, the different parameters are not always clearly separated or distinguished in the literature, and even where they are, it is frequently not clearly differentiated between the parameters when it comes to the analysis of trends (e.g. in studies which record both abundance and species richness and changes in these parameters, those parameters are usually still jointly discussed with regard to causative factors, without further differentiating between them). And second, breaking down our analysis according to individual measured parameters and only jointly evaluate e.g. studies which measure biomasses would certainly be the most accurate approach, but it would, considering the overall limited number of publications reporting long-term trends, lead to a very low number of studies per comparison, which would no more allow for a scientifically robust analysis. Therefore, yet being conscious of the limitations, we have decided to pursue an approach which is integrating the analysis over different parameters, bearing in mind that in many, yet not all cases the identified drivers and pressures will in fact influence different parameters in a similar direction.

### Trends

The total number of extracted long-term trends shows for both species groups that declines were more frequent than increases, which supports an overall declining trend for these taxa.

However, it should be kept in mind that a typical pitfall for any reviews is publication bias, meaning that 'positive' results, i.e. declines in the case of insect population trends, are more likely to be considered for submission for publication [31, 47]. Another potential limitation to be mentioned in this context is the heterogeneity in the verification levels of the pressures, which is discussed in more detail under the section Scientific verification of the pressures. Declining trends in insect populations do not appear to be a new phenomenon; for instance, a pronounced decline of invertebrates in the agricultural landscape between the 1950s and the 1980s have been described [48]. In this context, timescale is important as trends and drivers are not necessarily always uniform, they can also change over time; for instance, [14] show first increasing, then decreasing trends in the biomass of British moths since the 1960s.

Our evaluation focuses on the drivers behind the trends, allowing the integration of different parameters and regional scales (from local to country-wide) as well as varying methodology. In this approach, an evaluation of trend sizes and the importance of trends across the different literature sources would be hard to realize. For example, some of these were only partial trends, e.g. only affecting defined ecological groups (for instance habitat generalists vs. habitat specialists), or a few species favoured by climate change may show an increasing trend, whilst the overall group is declining (e.g. [40]). Therefore, all trends of species groups within an evaluated publication, regardless of their strength, were included in the evaluations.

## Drivers, pressures & states

Our findings suggests that whether reported trends are negative or positive, changes in insect populations have been primarily anthropogenically driven. The anthropogenic drivers include land use, agriculture, climate change (accelerated by human activities), nature conservation, urbanisation, and anthropogenic activities in general. In the last centuries, the changes resulting from human activities have accelerated (e.g. progressive industrialisation, intensive agriculture) and many species are unable to adapt to the conversion of habitats [49].

Anthropogenic activities, and in particular agricultural intensification, are the most discussed drivers of decreasing trends for carabids and lepidopterans. They drive pressures such as land use change and land consumption, management practices, eutrophication (fertilisers) as well as pesticides and pollution. This emphasis in long-term data is supported by shorter-term experimental studies [50–52] and correlations of different indicators for land use impact on biodiversity [53–55].

It is important to note that land use change and intensive land management practices are frequently closely interlinked with other pressures such as the use of fertilisers and pesticides [56, 57]. Intensive crop management practices, including the use of pesticides and fertilisers, are in many aspects inter-dependent with intensified land management and land use change. Some management practices are consequently fostering the use of agrochemicals or fertilisers, and vice versa. This complex subsequently influences habitat quality; for instance, reduction in weeds providing consistent food availability for flower-visiting lepidopterans or feeding plants of their larvae [58]. However, extensive cultivation can also lead to a decrease in species abundance but an increase in biodiversity [59]. Considering these dependencies, any assessment of biodiversity that involves habitat quality changes should be based on both qualitative (i.e. community structure, key species) and quantitative (i.e. species richness, individual numbers) parameters regarding the insect communities [19]. For records of numbers of individuals or species groups, without information about habitat quality or structure of the community, it is difficult or even impossible to address the specific trend as positive or negative. The crucial relevance of habitat quality is reflected in the results by the fact that it has a high impact (state) on the populations of both investigated species groups.

Climate and weather conditions in general, and climate change in particular, are important factors influencing insect populations and their development. These are complex phenomena which can manifest themselves in a broad range of parameters, e.g. increasing or decreasing temperatures or precipitation, extreme weather conditions, increasing climate variability, drought, inundations, changes in microclimate, and many more [60]. Accordingly, climate and weather interact with habitats and species, directly and indirectly, in various different ways. In recognition of this complexity, we have combined all climate-related pressures into a single "climate" pressure class. In our subsequent classification of drivers, we followed the authors of the original publications with their interpretation of climate data; where the climate was indicated as pressure, we assigned this to "climate change" as driver in cases where the authors classify it as climate change related, and to "natural drivers", where they classify it as climatic variability, or extreme weather events without making a link to climate change. The distinction between "climate change-related" and "weather-related," as equivalent to "not climate change-related," is debatable. However, it best reflects the analytic and interpretive depth of the original publications. While climate change is often cited with positive effects on insect populations, such as the geographical spread of species of warmer climatic zones [61–63], adverse effects are also reported for species that have more limited adaptability [64, 65]. Our analysis shows a similar pattern: climate change is cited as one of the top three pressures driving both increasing and decreasing trends for insect populations of carabids and lepidopterans. It mainly affects insect populations by changed climatic habitat conditions like humidity, microclimate-cooling effects or drought events [66–68]. Although there are reports for some butterfly species being potentially able to extend their distribution range due to climate warming, some of their populations are not only hampered by other pressures, e.g. landscape fragmentation, but could even be threatened with extinction if their current habitat undergoes further modification due to climatic changes [69, 70]. The example underlines the importance to consider multifactorial inter-relations of potentially interacting environmental pressures on biodiversity.

A review of publications related to climate change vulnerability of insect populations between 1991 and 2012 showed that the research field of climate change effects has grown from a handful of publications to more than 600 publications per year in the last decade [71]. In some of the publications that we analysed, a similar trend became evident as many authors either assessed or used publicly available weather data and climate models [72, 73] to correlate new and already published monitoring data of insect populations. Over both taxa investigated and irrespective of the trend types, a relatively small share of climate change indications was only presumed, in contrast to other pressures. On the one hand, this underlines the fact that where systematic data on environmental pressures are broadly available, research gaps can be addressed still years after the original insect monitoring data had been generated. On the other hand, it shows how funding, in this case done for climate change research in the last decades, can influence the direction of research and thereby the awareness of pressures in the current discussion.

Nature conservation measures are a primary driver for increasing trends of carabids and lepidopterans. Nature conservation activities take effect mainly through a change of cultivating land or by the restoration of habitats that improve the habitat quality and availability for species [20, 21]. Measures of nature conservation are related or respond to a diverse number of pressures like land use, forestation, management practice, or pollution. In order to address the agricultural intensification as a driver of the declines identified, i.e. to ensure a sustainable, insect diverse and productive agrarian landscape, nature conservation and productive agriculture would need to balance in order to maintain or enhance biodiversity to an appropriate level [21].

Looking at the states of the drivers of increase or decrease, it is conspicuous that the vast majority of the indications are attributed to habitat condition and habitat availability, some to habitat and organisms, and only a minority to the organisms directly. This underpins the pivotal role of the habitat in the current discussion and for the toolbox of measures to counteract insect decline. Habitat conservation and restoration may need to become a key concern in the future discussion to protect insect biodiversity on a broader scale. The category 'habitat & organisms' with a share of 27% illustrates that effects are often diverse, and organisms and habitats are both affected by some pressures or effects could not be separately attributed. Within the evaluated publications, effects that may be expected to have a direct impact on organisms without affecting habitats, were rarely cited.

## Scientific verification of the pressures

When classifying the pressures extracted from scientific literature, it became obvious that the potential causes of declines were supported by data of diverse quality. In the case of 'pressure presumed', there is a broad range of qualities of underlying data. This classification covers the entire spectrum from speculation to a detailed elaboration of causes that were, for various reasons, not examined in the evaluated publication but might have been well-examined elsewhere. Therefore, 'presumed' does not necessarily mean that there would be no evidence *per se*, but that there is no evidence-based elaboration of causes for the observed trend in the analysed study.

When comparing decreasing with increasing trends, it is interesting to note that 50–70% of the reasons cited for decreasing trends are identified on the basis of presumption. On the other hand, the pressures behind increasing trends were better examined, especially in the case of carabids. Here, the underlying hypotheses are possibly better defined, as many publications were reporting the results of efficiency control studies for ecological enhancement measures (e.g. investigation after habitat restoration or altered management practices). In these cases, it is easily possible to measure biodiversity parameters before and after the changes take place, and the local conditions can be assessed in detail. In contrast, decreasing trends may lack a similarly detailed hypothesis as a decrease are rarely intentionally induced or systematically observed from its onset onwards. Therefore, the hypothesis formation often only begins after the declining trend has been identified.

Furthermore, our results show how knowledge of ecological traits of species, especially in lepidopterans, can directly help to interpret the results of population trends. In some studies, the link between butterfly population trends and pressures such as eutrophication and associated changes in plant communities and host plant quality was so apparent that there was no need to systematically analyse it in the study. Therefore, in some cases presumed pressures can also be attributed to the level of knowledge about the ecology of the species investigated. Nevertheless, quantitative and statistically supported data or reliable correlations, e.g. of developed indicators like land use or fragmentation intensity, would lead to a higher certainty and possibly new insights concerning the pressures not only for Lepidoptera, but also for other taxa.

## Discussion of methods

A rather simplistic view on pressures and their aggregation and interpretation was criticised in recent publications addressing the discussions on the causes of insect decline [8, 24, 31, 74]. We employed the DPSIR model [35] to provide a systematic overview of interacting pressures and to elaborate on the major drivers. The corresponding uniform classifications enabled the precise allocation of local pressure to the appropriate fit on the different levels of the model like their underlying driver or following state. For example, the pressure "grazing" belongs to

the category "management practice". Management practice then is often associated with agricultural intensification. Grazing, however, was mentioned in different contexts as it occurs in agricultural intensification as well as in activities for nature conservation [1, 75]. In these cases, the DPSIR model allowed us to maintain the specific allocation even though we were using management practice as a more simplified pressure classification for the evaluation. In addition, these pressure classifications provided a better clarity and the possibility of a graphical representation (alluvial diagrams) of frequencies of the naming of individual pressures by the authors of the publications evaluated in the present analysis.

Our evaluation provides an overview of the frequency with which certain pressures are referred to in the scientific literature to explain population changes in the investigated insect groups, but inherently does not allow a quantification of the relevance of pressures in absolute figures, or a relative ranking of them according to their importance. One of the reasons for this is that there may be a bias in the investigation of pressures for a part of those studies in which specific examinations were made. For instance, it appears likely that pressures that can be readily quantified and for which precise figures are accessible, like for instance changes of local climate or land use, are better documented and thus more easily used as input parameters in a correlation or causal analysis. Therefore, studies looking at pressures which are difficult to quantify or are more complex, like agricultural practices or isolation of habitats, may be under-represented in the scientific literature.

Our approach was based on analysis of the conclusions drawn by the authors in the evaluated studies regarding the pressures contributing to observed population changes, and the importance of these pressures. This ensured a full representation of all extracted data sets, including both examined pressures and other pressures that authors referred to, but which were not necessarily analysed in the respective study. Long-term monitoring studies, however, are frequently designed to document trends of populations and biodiversity over time, but not necessarily to identify all related causes. Consequently, the respective studies do not necessarily allow the deduction of conclusions about the relative importance or scale of impact of the suggested causal factors. They rather represent the different facets of the complex inter-relationships of stressors. As an example, nature conservation can have a positive effect on biodiversity, but at the same time reduce the number of specific species e.g. the removal of willow trees for the maintenance of meadow habitats inevitably leads to the reduction of organisms dependent on willows.

As mentioned above, some of the publications did not aim for a detailed evaluation of causes behind the reported change. The authors of these publications did not give any further details about the driving forces behind the pressures, or they listed general complexes of potential causes without further differentiation. This issue is exemplified in the cases of land use, where 30 out of 150 indications, or of eutrophication, where 25 out of 40 indications are categorized as 'in general'. In such cases, it is difficult to retrospectively determine which exact pressure, such as fertiliser use or atmospheric deposition of nitrogen, contributed to the described eutrophication at the local level.

## Knowledge gaps and future research needs

To improve the scientific verification level of insect population trends in the future, programs should be designed to include monitoring of possible pressures alongside strategic monitoring of the selected taxonomic groups [76]. Both aspects are crucial for the interpretation of the resulting datasets. Moreover, to analyse the development of biodiversity over time and to assess changes, it is important to consider historical data like some of those analysed in this review [27]. Only by comparison of historical with recent data, the scale of change in terms of

quality and quantity can be assessed, and typical pitfalls of missing out gradual shifts due to a different perception over generations can be avoided (shifting baseline syndrome [77]). In this context, it is important to do a thorough statistical evaluation of observed trends and potential pressures and drivers wherever possible, in order to ensure a good verification level, avoiding presumptions and speculations about potential influencing factors.

For future discourse, in our opinion, the following questions are of great importance and should be answered stepwise by the scientific community and society:

- How can typical insect coenoses be defined in structure and abundances for different landscapes and habitats; where are their tipping points i.e. the point where the sum of different pressures leads to significant negative impact on the insect diversity and associated ecosystem services—and at what point of the dynamic are corresponding insect coenoses standing today?

- What kind of habitat network and habitat quality is necessary to ensure stable insect populations in anthropogenically influenced, especially agricultural landscapes, in the future?

- Which habitats and inhabiting insect species should be given priority in terms of conservation measures? I.e. which types of ecosystems are unique within Europe? How to define and determine them?

- How to reach a sustainable equilibrium of societal interests (such as food production) and nature conservation in human landscape management (e.g. agricultural practices) in terms of a dynamic process.

- How can those requirements be integrated into current discussions to build a coherent Trans-European Nature Network [78]?

Some answers about main drivers and pressures and therewith the direction of recommended action can already be given. However, to face the questions above in an appropriate manner, there is a strong need to assess the insect populations and their possible pressures more systematically in the future. It is crucial for legislation to set standards and thresholds that should be developed on a scientific basis to guide policymakers in the direction of a more sustainable land use system integrating societal, economic and nature conservation aspects.

## Supporting information

**S1 Table. Developed categories with lists of keywords in English and German.**
(PDF)

**S2 Table. Description of drivers, pressure class and state categories, obtained from the evaluated literature.** (see S4 Table).
(PDF)

**S3 Table. Pressures, pressure classes, drivers, states, obtained from the evaluated literature.** (see S4 Table).
(XLSX)

**S4 Table. Full list of analysed publications.**
(PDF)

**S5 Table. Full data base.**
(XLSX)

**S1 Fig. Analysed publications and their recording timespan and recording intervals, respectively.** (a) overview of publications per country in the focus region, (b) 33 publications with Carabidae trends, (c) 54 publications with Lepidoptera trends.
(PDF)

## Acknowledgments

The authors would like to thank Dr. Helen Thompson (Syngenta) for valuable remarks and a thorough linguistic revision of the English text of the manuscript.

## Author Contributions

**Conceptualization:** Quintana Rumohr, Christian Ulrich Baden, Matthias Bergtold, Michael Thomas Marx, Johanna Oellers, Michael Schade, Andreas Toschki, Christian Maus.

**Data curation:** Quintana Rumohr, Johanna Oellers, Andreas Toschki.

**Formal analysis:** Quintana Rumohr, Johanna Oellers, Andreas Toschki.

**Funding acquisition:** Christian Ulrich Baden, Matthias Bergtold, Michael Thomas Marx, Michael Schade, Christian Maus.

**Investigation:** Quintana Rumohr, Johanna Oellers, Andreas Toschki.

**Methodology:** Quintana Rumohr, Christian Ulrich Baden, Matthias Bergtold, Michael Thomas Marx, Johanna Oellers, Michael Schade, Andreas Toschki, Christian Maus.

**Project administration:** Quintana Rumohr, Johanna Oellers, Andreas Toschki.

**Resources:** Quintana Rumohr, Johanna Oellers, Andreas Toschki.

**Software:** Quintana Rumohr, Johanna Oellers, Andreas Toschki.

**Supervision:** Andreas Toschki, Christian Maus.

**Validation:** Quintana Rumohr, Christian Ulrich Baden, Matthias Bergtold, Michael Thomas Marx, Johanna Oellers, Michael Schade, Andreas Toschki, Christian Maus.

**Visualization:** Quintana Rumohr, Christian Ulrich Baden, Johanna Oellers, Andreas Toschki.

**Writing – original draft:** Quintana Rumohr, Johanna Oellers, Christian Maus.

**Writing – review & editing:** Quintana Rumohr, Christian Ulrich Baden, Matthias Bergtold, Michael Thomas Marx, Johanna Oellers, Michael Schade, Andreas Toschki, Christian Maus.

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
