## [Decision Letter · Decision Letter 0]

14 Mar 2023

PONE-D-23-00796Insect decline in countries of Central and Western Europe - Drivers and pressures of Carabidae and Lepidoptera population and biodiversity trendsPLOS ONE

Dear Dr. Baden,

Thank you for submitting your manuscript to PLOS ONE. After careful consideration, we feel that it has merit but does not fully meet PLOS ONE’s publication criteria as it currently stands. Therefore, we invite you to submit a revised version of the manuscript that addresses the points raised during the review process.

This manuscript has been now evaluated by two expert reviewers. Both of them find the work valuable and interesting, but have several concerns with it current presentation. Reviewer 1 requires higher clarification regarding the approach, the drivers and pressures. Reviewer 2 thinks that it should be clearer that this review focuses on agricultural landscapes, and that the discussion is quite long with much of the text focusing on ‘data limitation, quality etc’, whereas the interpretation of the findings should come more to the forefront. I concur with these comments and therefore invite the authors to conduct a deep revision of the manuscript. If the authors are interested in revising the manuscript, they should adjunct to the revised manuscript a response letter in which all the comments of the reviewers have been answered point-by-point.

We look forward to receiving your revised manuscript.

Kind regards,

Amparo Lázaro, PhD

Academic Editor

PLOS ONE

“CUB, MTM and CM are employees of BAYER AG; MB is employee of BASF SE and MS is employee of Syngenta Group Co., Ltd.

www.bayer.com

www.basf.com

www.syngenta.com”

Reviewers' comments:

Reviewer's Responses to Questions

**Comments to the Author**

1. Is the manuscript technically sound, and do the data support the conclusions?

Reviewer #1: Yes

Reviewer #2: Yes

2. Has the statistical analysis been performed appropriately and rigorously? 

Reviewer #1: N/A

Reviewer #2: N/A

3. Have the authors made all data underlying the findings in their manuscript fully available?

Reviewer #1: Yes

Reviewer #2: Yes

4. Is the manuscript presented in an intelligible fashion and written in standard English?

Reviewer #1: Yes

Reviewer #2: Yes

5. Review Comments to the Author

Reviewer #1: I carefully read the manuscript “Insect decline in countries of Central and Western Europe - Drivers and pressures of Carabidae and Lepidoptera population and biodiversity trends”. The manuscript presents an exhaustive review on “The drivers and pressure behind insect decline based on long term monitoring data” of a specific region of Europe.

Although there are many studies reporting no significant changes in insect numbers, there is clear a significant bias against publishing papers about non-significant findings. Therefore, as the authors discussed, it is difficult to evaluate insects’ trends based on literature review. However, this work is more related to the identify the drivers and pressures reported in the literature to the different reported trends (i.e. decreasing vs. increasing). The authors used a useful a conceptual model (DPSIR) to disentangle several drivers and pressures influencing different states (insect individuals, populations and communities and their habitats). I enjoyed reading this well-written manuscript integrating data of an important amount of long-term studies and very illustrative figures on the pressures and drivers behind insect trends.

Major comments:

1-The title of the paper specifies that the work is related to “population and biodiversity trends” but in different parts of the manuscript it seems that the analyses are only related to evaluating population trends (Abstract L. 37-39; Introduction L. 84-87 and L. 91-92; Discussion L. 391). However, it is clear that they evaluated several parameters which do not measure population change (e.g. species richness, diversity indices, distribution). My main concern on the authors’ approach is analysing together the drivers and pressures of the trends in parameters measuring many different ecological facets of populations and insect communities. Although the approach could be useful to identify the major reported drivers and pressures of insect biodiversity trends, analysing together all these parameters difficult the possibility of evaluating their real impact on populations and communities. Clearly, in a specific location, contrasting trends could occur with respect to different parameters (e.g. increase in insect abundance but a decrease in insect diversity or increase in range size distribution of a species but a decrease in the population sizes). In any case, I think a discussion on the evaluated parameters and how they measure insect biodiversity should be incorporated in the manuscript. Also, I would like to see a justification of why differentiating between abundance measured by indices and “number of individuals”. It seems weird to use the term DPSIR model when the authors did not evaluate the “Impact” neither the “Response”. I may incorporate in the model and the Figure 1 which facet of insect biodiversity is the pressure affecting (which is measured by the different parameters). In fact, Table S5 report the trends separated by the different parameters. I may include this table in the manuscript.

2-As the authors mention in the manuscript, climate change is one of the most studied drivers to interpret insect’s trends and can imply several pressures to insects but only climate warming was included in Figure 1 (and climate change is not included as a driver in the figure). Therefore, I would like to see which specific pressures driven by climate change (i.e climate change in general in Table S3 = 38 studies) have been evaluated in the study region, including climate extremes, increased climate variability, water stress and which of them have been reported to have an impact on the habitat and/or on the organisms.

3-The pressure “weather/climate in general” (n=19) was classified as “climate” together with climate change pressures (“climate change in general”, “climate cooling” and “climate warming”). I expect that inter-annual weather/climate variability to be responsible of population fluctuations in insect but climate change to drive the long-term trends. Then, it seems incorrect to place in the same group (even including it as a pressure) the natural climate variability and the climate change pressures (e.g. long-term warming trends). I do not see in Table S4 which studies have analysed or mentioned the different pressures or which parameters they measured.

4- I would like to see in the introduction or the methods a paragraph about the study region. Why is this region interesting for evaluating insect trends (e.g. more data than other regions)? Why could be the particularities of the situation of insects in the study region? I expect that countries of Central and Western Europe with high population densities (e.g. compared to USA or Canada) are highly impacted by human activities (e.g. agriculture).

Minor comments:

Abstract:

- L. 19. I may say here that most of the long-term reported trends are from regions of USA and Europe. Now it could seem that these trends have been reported only in specific regions of Europe.

-L. 27. I may include the name of the order (Coleoptera) after Carabidae.

-L. 30. Change “environmental stressors” by “related factors”.

L. 36-37. I’m not sure here if I understand what do you refer for “intrinsic changed related to insect species or the environment”. Could you clarify it?

Introduction:

-L. 75. How a stressor could reduce competition? Do you mean that a stressor reducing the population numbers of some taxa decrease the competition with other taxa which can their populations?

Methods:

- L. 190. I do not like the use of “organism” as a synonym of “specie”s or vice versa.

Results:

L. 235-236. Could you include the number of publications by country?

L. 373. I’m surprised that forestation led to increasing biodiversity trends in Lepidoptera in the study region. At least in the Mediterranean region, there are some examples that the ecological succession from grasslands to scrublands and forests are related to declining trends in Lepidoptera. I guess forestation in degraded habitat could have a benefit on Lepidoptera? Could you tell me which are the publications reporting the positive effect of forestation in insect trends?

Discussion:

L. 494-495. Species richness (i.e. number of species) is a quantitative parameter.

Figures and tables:

-Figure 1: I’m not an expert on ecological resilience but I think this concept is more related to communities or populations than to organisms. In L. 165 it is mentioned “resilience of species”.

-Table S3: A more detailed legend is required.

-Table S4. Include pressures tested or mentioned.

Reviewer #2: Review: Rumohr et al. ‘Insect decline in countries of Central and Western Europe – Drivers and pressures of Carabidae and Lepidoptera population and biodiversity trends, to be considered for publication in the journal PLoS ONE.

Reviewer’s interpretation of the work: The authors performed a literature review on carabid beetle and butterfly trends (datasets of at least 6 years) in agricultural landscapes of south and western Europe. They identified increasing/stable/decreasing/unknown trends in various parameters (abundance, richness, etc.) and linked these to drivers, pressures and states (using the DPSIR methodology and presenting alluvial diagrams). Trends were mostly negative (i.e., a decrease in the insect parameter), but positive trends were also observed. Drivers were primarily anthropogenic, primarily agriculture, climate, nature conservation activities, urbanisation, etc. Results are discussed in detail, also with a focus on data quality and potential pitfalls in their study.

Specific comments:

The title is pretty long, and it seems that there is even a suggested ‘subtitle’. Why not simply use a modified subtitle as the main title, and remove the main one: Drivers and pressure behind insect decline in western Europe based on long-term monitoring data

L25. ‘drivers and pressures’ Later on in the article, these terms were described to have different levels of confidence, depending on whether they were tested or inferred. Make this clear in the Abstract.

Since Carabidae and Lepidoptera are at different taxonomic levels, I would state here (but no need to do this throughout the manuscript): Carabidae (Coleoptera) and Lepidoptera…

L28. ‘…data for at least 6 years…’ I presume (based on what comes later) that this is not 6 consecutive years, but rather data spanning at least 6 years, correct? For instance, could the data include a study that was performed in year X and then only again in year X + 6?. Also, later on the authors told us that they focused their work on agricultural landscapes (L87 & Table S1). This needs to be made clear in the Abstract.

L46-7. Another citation that can be used here is:

Declining diversity and abundance of High Arctic fly assemblages over two decades of rapid climate warming

Loboda, et al. Ecography https://doi.org/10.1111/ecog.02747

To be fair, there are also opposite findings (or at least findings of no change), that the readers should be made aware of:

No net insect abundance and diversity declines across US Long Term Ecological Research sites

Crossley et al. Nature Ecology & Evolution volume 4, pages 1368–1376 (2020)

L89-95. In the sentences above the objectives, the authors emphasised that the work is performed in the agricultural landscape of C and W Europe, but the objectives do not emphasise this. It definitely should come to the forefront here.

L116. I may have missed it, but what is the timespan of your literature search?

L130-1. Were the Carabidae also evaluated ‘in general’ and as specialists, generalists, etc.?

Table S2, S3 – state that these were obtained from your literature review, referring to Table S4

Table 1. Provide a more descriptive title. For instance, what do you mean by ‘pressures discussed in total’, ‘important pressures discussed in total’? Do you mean total amount of pressures/important pressures discussed in the papers?

Table S5 seems quite important/interesting. I would present it in the text, rather than the supplementary material.

L266-8. I find this sentence difficult to understand. Please rewrite.

L280. No need to cite Figure 2a an b here. You’re talking about alluvial diagrams in general. Or you could simply state Figs. 2a, 3a, 4a, 5a, and then Figs. 2b, 3b, 4b, 5b.

L286 onwards. Carabidae decreasing/increasing trends & Lepidoptera decreasing/increasing trends. There’s a lot to digest here. I recommend the authors provide a summary paragraph above these sections, summarising the main findings. Perhaps dealing with decreasing trends first (both taxa), and then increasing trends (both taxa). Then go into the nitty-gritty of each of the 4 sections.

L336-9. It is fascinating that for increasing trends (carabids), there is considerable statistical evidence, while for decreasing trends (carabids), there is so little (and mainly presumed). Any reason for this huge discrepancy? I see that this is discussed in the Discussion.

Figure quality should be improved.

Discussion. Make it clear early on that this study focused on agricultural landscapes. It may not be surprising that agricultural activity/intensification/extensification were often an important driver in the trends observed. In other landscapes (semi natural to natural, urban, etc.) there may be different prominent drivers of insect decline/increase.

L396 onwards. Apart from publication bias, the authors also need to discuss the confidence in these trends, since many were not supported by statistical analyses but were presumed. I see this is done later on in the Discussion, but perhaps also briefly mention it here.

“Discussion of methods”. I recommend that this section be placed at the end of the Discussion, perhaps together with “Knowledge gaps and future research needs”. The reader might be more interested to read the interpretations of your findings first, and then later on be made aware of the potential methods issues/pitfalls and knowledge gaps, etc.

L428-436. Not sure if this Google scholar vs. other courses text is needed. Perhaps some of this is common knowledge?

L439-49. The text here can go with the “Discussion of methods” etc. (see my comment above on this). At this stage, the reader wants to read the interpretation of your findings. In fact, so too are the next two paragraphs. The interpretation of your findings mainly starts in L471.

Knowledge gaps, etc. An additional action would be to have more statistical evidence of the trends observed. As the authors have (L555 onwards), much of this is based on ‘presumption’.

6. PLOS authors have the option to publish the peer review history of their article (what does this mean?). If published, this will include your full peer review and any attached files.

Reviewer #1: No

Reviewer #2: No

---

## [Author Response · Author response to Decision Letter 0]

28 Apr 2023

Dear Editor and Reviewers,

please find here our answers and comments embedded in the reviewers’ comments in red. We hope to have answered and addressed all points mentioned. We highly appreciate the time and effort you have spent with our manuscript, be assured that it got way more sophisticated thanks to your help.

We adapted our financial disclosure as well:

„CUB, MTM, CM, MB, and MS are co-authors and contributed to the establishment of the study concept and to the evaluation and interpretation of the data, and to the preparation of the manuscript, as stated. The study was financially supported by Bayer, BASF, and Syngenta”.

Best regards

Christian Baden

Review Comments to the Author

Reviewer #1: 

I carefully read the manuscript “Insect decline in countries of Central and Western Europe - Drivers and pressures of Carabidae and Lepidoptera population and biodiversity trends”. The manuscript presents an exhaustive review on “The drivers and pressure behind insect decline based on long term monitoring data” of a specific region of Europe.

Although there are many studies reporting no significant changes in insect numbers, there is clear a significant bias against publishing papers about non-significant findings. Therefore, as the authors discussed, it is difficult to evaluate insects’ trends based on literature review. However, this work is more related to the identify the drivers and pressures reported in the literature to the different reported trends (i.e. decreasing vs. increasing). The authors used a useful a conceptual model (DPSIR) to disentangle several drivers and pressures influencing different states (insect individuals, populations and communities and their habitats). I enjoyed reading this well-written manuscript integrating data of an important amount of long-term studies and very illustrative figures on the pressures and drivers behind insect trends.

The authors would like to thank Reviewer #1 for his / her positive feedback and for his / her valuable remarks and suggestions which helped us a lot to improve the manuscript. In the following, please find an outline how we have addressed the individual comments and implemented the respective adaptations of the manuscript.

Major comments:

1-The title of the paper specifies that the work is related to “population and biodiversity trends” but in different parts of the manuscript it seems that the analyses are only related to evaluating population trends (Abstract L. 37-39; Introduction L. 84-87 and L. 91-92; Discussion L. 391). However, it is clear that they evaluated several parameters which do not measure population change (e.g. species richness, diversity indices, distribution). My main concern on the authors’ approach is analysing together the drivers and pressures of the trends in parameters measuring many different ecological facets of populations and insect communities. Although the approach could be useful to identify the major reported drivers and pressures of insect biodiversity trends, analysing together all these parameters difficult the possibility of evaluating their real impact on populations and communities. Clearly, in a specific location, contrasting trends could occur with respect to different parameters (e.g. increase in insect abundance but a decrease in insect diversity or increase in range size distribution of a species but a decrease in the population sizes). In any case, I think a discussion on the evaluated parameters and how they measure insect biodiversity should be incorporated in the manuscript. 

This is a very valid and important point, thanks for flagging it up. We have taken this matter into consideration when establishing the concept of our study - indeed, it would be most accurate to only jointly evaluate data sets which analyze the same sort of parameters (i.e. only to include studies in the same evaluation which look at changes e.g. in biomass, and not species richness). However, there are two, predominantly practical main reasons why we had decided to pursue an alternative approach instead, i.e. to integrate over different parameters in the same analysis: first, in the original papers, the different parameters are frequently not clearly differentiated when it comes to the analysis of factors driving the observed trends (e.g. a study which is describing changes in abundance and species richness will normally not analyze the drivers of changes in those two parameters separately), and second, restricting the analysis to studies where the same parameter has been assessed would, considering the overall limited number of long-term studies, have pushed the number of papers included in the analysis below the “critical mass” of data needed for a sound evaluation. We have summarized the rationale behind this in a new paragraph at the beginning of the discussion and adapted accordingly the sections where we were referring to population level changes only instead to changes in a broader context.

Also, I would like to see a justification of why differentiating between abundance measured by indices and “number of individuals”. 

This differentiation has been made since in some studies (especially on Lepidoptera) not individual numbers were indicated, but instead evaluations of abundances by means of indices were reported (e.g. TRIM (TRends and Indices for Monitoring data, Pannekoek and Van Strien, 2001) oder RRR (relative reporting rate). A note explaining this has been added to Materials & Methods, Overview of the Data Basis. We have outlined this under Results – Overview of the data basis.

It seems weird to use the term DPSIR model when the authors did not evaluate the “Impact” neither the “Response”. 

Most of the evaluated studies were not organized according to the DPSIR Model and did not take into consideration impact and response, so that we had to exclude these parameters, as the studied literature did not provide the respective information. We have tried to describe this more clearly in Materials & Methods under Evaluation of population trends - Drivers and pressures of the extracted trends.

I may incorporate in the model and the Figure 1 which facet of insect biodiversity is the pressure affecting (which is measured by the different parameters). In fact, Table S5 report the trends separated by the different parameters. 

See remark on Major comments #1

I may include this table in the manuscript. 

Table integrated in the MS

2-As the authors mention in the manuscript, climate change is one of the most studied drivers to interpret insect’s trends and can imply several pressures to insects but only climate warming was included in Figure 1 (and climate change is not included as a driver in the figure)

Changed into “climate” in Fig. 1 to address this point (furthermore, we should mention that the drivers and pressures shown in Fig. 1 are only examples). 

Therefore, I would like to see which specific pressures driven by climate change (i.e climate change in general in Table S3 = 38 studies) have been evaluated in the study region, including climate extremes, increased climate variability, water stress and which of them have been reported to have an impact on the habitat and/or on the organisms. 

In the evaluated literature, all kind of climatic factors were dealt with and discussed regarding their influence on insect populations and communities. On the level of the individual publications, these factors are shortly summarized in Table S5, column I. Due to the diversity of climate-related factors, and the fact that authors of the evaluated publications dealt with them in a very heterogenous way, it was not possible to individually classify them in more detail in categories for the evaluation. Therefore, we summarized climate and weather-related factors under “climate” on pressure class level, and either under “climate change” or under “natural drivers” on driver level, depending on whether the authors of the original publications assigned the factors to climate change or to other climate / weather parameters. To make this clearer, we have outlined these aspects in a new section in the Discussion under Drivers, Pressures & States.

3-The pressure “weather/climate in general” (n=19) was classified as “climate” together with climate change pressures (“climate change in general”, “climate cooling” and “climate warming”). I expect that inter-annual weather/climate variability to be responsible of population fluctuations in insect but climate change to drive the long-term trends. Then, it seems incorrect to place in the same group (even including it as a pressure) the natural climate variability and the climate change pressures (e.g. long-term warming trends). I do not see in Table S4 which studies have analysed or mentioned the different pressures or which parameters they measured. 

We agree that climate change, which entails profound long-term changes, is likely to influence insect populations and communities in a different way than “normal” climate and weather variability, extreme weather events, and changes in microclimate. In practice, however, it is frequently hard to clearly differentiate between weather and climate / climate change factors in the literature, first, since many studies do not allow for an unequivocal classification of the climate parameters they are describing, and second, since weather and climate / climate change factors are in practical instances frequently not straightforward to tell apart as they are interwoven in a complex way, and can influence and overlay each other, especially when longer time frames are taken into consideration as we do in this study. In our analysis, as a pragmatic solution, we therefore followed the authors of the original publications with their interpretation of climate data; where the climate was indicated as pressure, we assigned this to “climate change” as driver in cases where the authors classify it as climate-change related, and to “natural drivers”, where they classify it as climatic variability, microclimate, or extreme weather events without making a link to climate change. To make this clearer, we have outlined these aspects in a new section in the Discussion under Drivers, Pressures & States. The different climate and weather parameters studied in the individual publications are summarized in Table S5, column I.

4- I would like to see in the introduction or the methods a paragraph about the study region. Why is this region interesting for evaluating insect trends (e.g. more data than other regions)? Why could be the particularities of the situation of insects in the study region? I expect that countries of Central and Western Europe with high population densities (e.g. compared to USA or Canada) are highly impacted by human activities (e.g. agriculture). 

Rationale added to the Materials & Methods section. Main reasons are a relatively comparable set of environmental conditions in countries of the Central and Northwestern European climatic Zone which has been significantly shaped by agriculture, and the comparably good availability of entomofaunistic data. As a side remark, we did a similar exercise for North America and found that relevant data from this region are by far more scarce, so that a scientifically sound interpretation was hardly feasible there. 

Minor comments:

Abstract:

- L. 19. I may say here that most of the long-term reported trends are from regions of USA and Europe. Now it could seem that these trends have been reported only in specific regions of Europe. - 

Adapted

-L. 27. I may include the name of the order (Coleoptera) after Carabidae. 

Added

-L. 30. Change “environmental stressors” by “related factors”. - 

Adapted

L. 36-37. I’m not sure here if I understand what do you refer for “intrinsic changed related to insect species or the environment”. Could you clarify it? 

Addressed

Introduction:

-L. 75. How a stressor could reduce competition? Do you mean that a stressor reducing the population numbers of some taxa decrease the competition with other taxa which can their populations? 

Sentence re-worded to make it better understandable

Methods:

- L. 190. I do not like the use of “organism” as a synonym of “specie”s or vice versa. 

-> Adapted

Results:

L. 235-236. Could you include the number of publications by country? 

Information added for key countries; detailed information indicated in Figure S1a; reference to figure included in the text.

L. 373. I’m surprised that forestation led to increasing biodiversity trends in Lepidoptera in the study region. At least in the Mediterranean region, there are some examples that the ecological succession from grasslands to scrublands and forests are related to declining trends in Lepidoptera. I guess forestation in degraded habitat could have a benefit on Lepidoptera? Could you tell me which are the publications reporting the positive effect of forestation in insect trends?

The respective pulications are:

• Conrad et al. (2004) (Lepidoptera)

• Laussmann et al. (2010) (Lepidoptera)

• Dennis et al. (2019) (Lepidoptera)

Furthermore Desender et al. (1994) for Carabidae

Discussion:

L. 494-495. Species richness (i.e. number of species) is a quantitative parameter. 

-> Adapted

Figures and tables:

-Figure 1: I’m not an expert on ecological resilience but I think this concept is more related to communities or populations than to organisms. In L. 165 it is mentioned “resilience of species”. 

Changed into “resilience of populations” 

-Table S3: A more detailed legend is required. 

More detailed caption added.

-Table S4. Include pressures tested or mentioned. 

Information provided in the table S5

Reviewer #2: 

Review: Rumohr et al. ‘Insect decline in countries of Central and Western Europe – Drivers and pressures of Carabidae and Lepidoptera population and biodiversity trends, to be considered for publication in the journal PLoS ONE.

Reviewer’s interpretation of the work: The authors performed a literature review on carabid beetle and butterfly trends (datasets of at least 6 years) in agricultural landscapes of south and western Europe. They identified increasing/stable/decreasing/unknown trends in various parameters (abundance, richness, etc.) and linked these to drivers, pressures and states (using the DPSIR methodology and presenting alluvial diagrams). Trends were mostly negative (i.e., a decrease in the insect parameter), but positive trends were also observed. Drivers were primarily anthropogenic, primarily agriculture, climate, nature conservation activities, urbanisation, etc. Results are discussed in detail, also with a focus on data quality and potential pitfalls in their study.

The authors would like to thank Reviewer #2 for his / her positive feedback and for his / her valuable remarks and suggestions which helped us a lot to improve the manuscript. In the following, please find an outline how we have addressed the individual comments and implemented the respective adaptations of the manuscript.

Specific comments:

The title is pretty long, and it seems that there is even a suggested ‘subtitle’. Why not simply use a modified subtitle as the main title, and remove the main one: Drivers and pressure behind insect decline in western Europe based on long-term monitoring data 

-> Adapted

L25. ‘drivers and pressures’ Later on in the article, these terms were described to have different levels of confidence, depending on whether they were tested or inferred. Make this clear in the Abstract. 

->Abstract adapted accordingly

Since Carabidae and Lepidoptera are at different taxonomic levels, I would state here (but no need to do this throughout the manuscript): Carabidae (Coleoptera) and Lepidoptera… 

-> Added

L28. ‘…data for at least 6 years…’ I presume (based on what comes later) that this is not 6 consecutive years, but rather data spanning at least 6 years, correct? For instance, could the data include a study that was performed in year X and then only again in year X + 6?. Also, later on the authors told us that they focused their work on agricultural landscapes (L87 & Table S1). This needs to be made clear in the Abstract. 

Indeed, the six years did not necessarily have to be consecutive. We have clarified this in the Materials & Methods section, and included a note into the abstract about the focus on agricultural landscapes.

L46-7. Another citation that can be used here is:

Declining diversity and abundance of High Arctic fly assemblages over two decades of rapid climate warming

Loboda, et al. Ecography https://doi.org/10.1111/ecog.02747

-> Added

To be fair, there are also opposite findings (or at least findings of no change), that the readers should be made aware of:

No net insect abundance and diversity declines across US Long Term Ecological Research sites

Crossley et al. Nature Ecology & Evolution volume 4, pages 1368–1376 (2020) 

Added

L89-95. In the sentences above the objectives, the authors emphasised that the work is performed in the agricultural landscape of C and W Europe, but the objectives do not emphasise this. It definitely should come to the forefront here. 

-> Adapted accordingly in the text

L116. I may have missed it, but what is the timespan of your literature search? 

-> See Results: Overview of the analyzed publications

L130-1. Were the Carabidae also evaluated ‘in general’ and as specialists, generalists, etc.?

Yes. Added to the text.

Table S2, S3 – state that these were obtained from your literature review, referring to Table S4 � Added for Table S3 

Added to the captions of the tables.

Table 1. Provide a more descriptive title. For instance, what do you mean by ‘pressures discussed in total’, ‘important pressures discussed in total’? Do you mean total amount of pressures/important pressures discussed in the papers? 

Title adapted to better describe the content of the table.

Table S5 seems quite important/interesting. I would present it in the text, rather than the supplementary material. 

Integrated into the manuscript

L266-8. I find this sentence difficult to understand. Please rewrite. 

Re-phrased.

L280. No need to cite Figure 2a an b here. You’re talking about alluvial diagrams in general. Or you could simply state Figs. 2a, 3a, 4a, 5a, and then Figs. 2b, 3b, 4b, 5b. 

Adapted

L286 onwards. Carabidae decreasing/increasing trends & Lepidoptera decreasing/increasing trends. There’s a lot to digest here. I recommend the authors provide a summary paragraph above these sections, summarising the main findings. Perhaps dealing with decreasing trends first (both taxa), and then increasing trends (both taxa). Then go into the nitty-gritty of each of the 4 sections. 

Added

L336-9. It is fascinating that for increasing trends (carabids), there is considerable statistical evidence, while for decreasing trends (carabids), there is so little (and mainly presumed). Any reason for this huge discrepancy? I see that this is discussed in the Discussion. 

As the reviewer rightly points out, this is addressed in the discussion. 

Figure quality should be improved. 

New figures provided with better resolution.

Discussion. Make it clear early on that this study focused on agricultural landscapes. It may not be surprising that agricultural activity/intensification/extensification were often an important driver in the trends observed. In other landscapes (semi natural to natural, urban, etc.) there may be different prominent drivers of insect decline/increase. 

-> Added

L396 onwards. Apart from publication bias, the authors also need to discuss the confidence in these trends, since many were not supported by statistical analyses but were presumed. I see this is done later on in the Discussion, but perhaps also briefly mention it here. 

Added

“Discussion of methods”. I recommend that this section be placed at the end of the Discussion, perhaps together with “Knowledge gaps and future research needs”. The reader might be more interested to read the interpretations of your findings first, and then later on be made aware of the potential methods issues/pitfalls and knowledge gaps, etc. 

Paragraph shifted

L428-436. Not sure if this Google scholar vs. other courses text is needed. Perhaps some of this is common knowledge? 

Removed

L439-49. The text here can go with the “Discussion of methods” etc. (see my comment above on this). At this stage, the reader wants to read the interpretation of your findings. In fact, so too are the next two paragraphs. The interpretation of your findings mainly starts in L471. 

Paragraphs shifted

Knowledge gaps, etc. An additional action would be to have more statistical evidence of the trends observed. As the authors have (L555 onwards), much of this is based on ‘presumption’. 

Added

---

## [Decision Letter · Decision Letter 1]

21 Jul 2023

Drivers and pressures behind insect decline in Central and Western Europe based on long-term monitoring data

PONE-D-23-00796R1

Dear Dr. Baden,

We’re pleased to inform you that your manuscript has been judged scientifically suitable for publication and will be formally accepted for publication once it meets all outstanding technical requirements.

Kind regards,

Amparo Lázaro, PhD

Academic Editor

PLOS ONE

Additional Editor Comments (optional):

The reviewer suggested the relocation of the section entitled "Parameters measured" to the methods section. The authors may consider this suggestion; both locations could be adequate

Reviewers' comments:

Reviewer's Responses to Questions

**Comments to the Author**

1. If the authors have adequately addressed your comments raised in a previous round of review and you feel that this manuscript is now acceptable for publication, you may indicate that here to bypass the “Comments to the Author” section, enter your conflict of interest statement in the “Confidential to Editor” section, and submit your "Accept" recommendation.

Reviewer #1: All comments have been addressed

2. Is the manuscript technically sound, and do the data support the conclusions?

Reviewer #1: Yes

3. Has the statistical analysis been performed appropriately and rigorously? 

Reviewer #1: N/A

4. Have the authors made all data underlying the findings in their manuscript fully available?

Reviewer #1: Yes

5. Is the manuscript presented in an intelligible fashion and written in standard English?

Reviewer #1: Yes

6. Review Comments to the Author

Reviewer #1: I appreciate the effort made by the authors for improving the manuscript following my suggestions and adresing my inquires. At this step I only would like to propose relocating the section titled "Parameters measured" (or potentially renamed as "Community and population parameters considered") to the methods section where it would be more contextually suitable.

7. PLOS authors have the option to publish the peer review history of their article (what does this mean?). If published, this will include your full peer review and any attached files.

Reviewer #1: **Yes: **Pau Colom

---

## [Editor Report · Acceptance letter]

27 Jul 2023

PONE-D-23-00796R1 

Drivers and pressures behind insect decline in Central and Western Europe based on long-term monitoring data 

Dear Dr. Baden:

I'm pleased to inform you that your manuscript has been deemed suitable for publication in PLOS ONE. Congratulations! Your manuscript is now with our production department. 

Kind regards, 

on behalf of

Dr. Amparo Lázaro 

Academic Editor

PLOS ONE